# A New Approach to Assess the Retained Value of Functionalized and Stabilized Wood Products through Aging

Diego Elustondo *[ID] and Douglas Gaunt

New Zealand Forest Research Institute Limited (Scion), 49 Sala Street, Private Bag 3020,
Rotorua 3046, New Zealand; douglas.gaunt@scionresearch.com
* Correspondence: diego.elustondo@scionresearch.com; Tel.: +64-7-343-5899

**Abstract:** Wood stabilization and functionalization is a hot topic towards increasing the use of wood in buildings. Building construction and materials account for around 11% of the global $CO_2$ emissions, thus there is increasing interest in using wood to replace concrete, steel, and aluminium. However, the perceived quality of wood in service decreases quickly in comparison with non-biodegradable alternatives, so aging must be delayed as much as possible through stabilization and functionalization. The question addressed in this paper is how to measure the perceived quality of wood products in service. The concept of wood quality is difficult to define, as it depends on a combination of factors such as customer satisfaction, wood fibre characteristics, products, applications, and so on. This paper proposes a definition of timber quality based on market price. By knowing the market value of a potential range of wood products, the proposed method calculates the loss of value due to developing defects caused by aging. Overall, the proposed methodology allows converting the subjective concept of aging into an objective $ value. A numeric example is presented based on the New Zealand timber grading rules. The results showed that 5.1 m by 150 mm by 50 mm softwood timber can lose up to 61% of its value as appearance grade if a single aging defect develops beyond the maximum allowed size.

**Keywords:** wood stabilization; wood quality; wood defects; wood aging; loss of value; New Zealand timber grading rules





## 1. Introduction

It has been estimated that buildings are responsible for around 40% of the global $CO_2$ emissions. Building materials and construction (also known as embodied carbon) are responsible for 11% of those emissions [1]. Since concrete, steel, and aluminium are very common energy-intensive building materials, replacing such materials with wood will support a transition to low emission and climate-resilient economy. If the predictions are accurate, almost 70% of the world's population will live in urban areas by 2050 [2], and wood materials will be instrumental to provide housing needs.

The environmental benefits of using engineered wood products in house construction have been demonstrated extensively [3]. Those benefits can go even further if environmental assessments include sustainability factors such as wellbeing, planet, and prosperity [4]. However, this unique opportunity will not be realized if the wood products industry continues business as usual. Exposed wood in building envelopes must be recoated between 4 and 12 times over a 50-year service life [5].

Furthermore, the B2 Durability clause in the New Zealand building code [6] stipulates that building envelope elements with normal maintenance only require to perform 15 years under specifications, but customers and designers usually expect much more. This can make maintenance prohibitive for many multi-storey buildings and multi-family dwellings.

This is where wood modification will play a crucial role, as builder developers will likely prefer materials that can retain their value during service life. The literature indicates

that wood modification started in the first half of the 20th century due to restrictions on biocides and the use of tropical hardwoods [7]. The scientific literature compiles a wide range of chemical, thermal, surface and impregnation methods that has been proposed in the past [8], and some of them, such as acetylation, furfurylation, and thermal modification, have reached commercialization [9]. More recently, researchers also investigated the use of nanotechnology for wood modification and protection [10,11].

Although there is not general rule, the term "wood modification" is applied to methods that change material properties, while the term "wood protection" is applied to methods that increase resistance to decay by organisms such as fungi, insects, bacteria, and marine borers. Wood products have been traditionally protected with chromated copper arsenate [12], and there are standards on how to measure wood durability by assessing decay after a number of years [13]. The data shows that service life above ground is between 3.5 and 9.5 years for reference species depending on the site [14], but it can be extended to more than 50 or 60 years by treating with creosote and copper formulations [15]. There are currently models in the literature for predicting decay rates in above-ground applications [16].

However, even though wood protection can delay decay for decades, it does not necessarily preserve appearance. Untreated wood is an appealing material that enhances the built environment with aesthetic sophistication and connection to nature, but this intrinsic value is quickly lost when untreated wood is exposed to the external environment. In the wood science jargon, losing any aspect of quality during time is referred as weathering [17] or aging [18]. For example, wood colour is expected to fade with exposure to sunlight [19], sometimes after a few months in service [17], and despite considerably extensive attempts the research towards developing durable clear coatings for exterior wood has virtually failed [20].

Consumers are generally aware that colour will fade, but colour changes do not typically occur uniformly in aged wood. Other common symptoms of aging, such as distortion and checks, are also perceived as quality defects. It has been shown for example that unpainted spruce and pine surfaces can degrade rapidly when exposed to the weather [17]. After 1 month the paint-holding properties of the wood are adversely affected and continue to deteriorate throughout the next 6 to 10 months.

Wood modification can prevent or delay aging, but the wood material will have higher initial cost that needs to be compared with the retained value. This paper addresses the question of how to assess the retained value of functionalized and stabilized wood products through aging. The objective is to propose an objective measure of wood quality, so that the higher price of a quality wood product can be compared with the lower loss of value in service.

According to the literature, the word "quality" is often used with affirmative and privative meanings [21]. Quality can be described as the presence of positive attributes, such as durability and dimensional stability, or the absence of negative attributes, such as not having knots, stains, and checks. Consequently, the meaning of quality in wood products will depend on customer expectations in comparison with the alternatives.

Researchers in the past tried to quantify quality through questionnaires. Questions were grouped in subjects describing different aspects of the concept of quality, so that the relative importance of each subject could be compared statistically. For example, one study [22] grouped questions related to performance, features, reliability, conformance, durability, serviceability, aesthetics, and supplier reputation. Another study [23] employed statistical analysis to assess the relative effects of different subjects and concluded that supplier services, wood performance and wood appearance had the higher impact on quality.

However, assessing wood quality with questionnaires is subjective. For example, researchers in the past noticed that grading rules for structural timber did not correspond well with customer perception [23,24]. For example, when home builders in New Zealand were asked to classify samples of timber according to distortion, they rejected timber with

crook and bow higher than 5 mm approximately [25]. In practice, the New Zealand timber grading rules [26] allow up to 10 mm crook and 15 mm bow for the same commercial product.

Researchers in the past have also tried to predict wood quality through physical and chemical properties that affect product performance in service [27,28]. For example, the effect of annual ring orientation [29], juvenile wood [30], and spiral grain [31] in wood dimensional stability is well documented. Wood features that have been associated to wood quality in the past are density, stiffness, strength, shrinkage, the proportion of juvenile, heartwood and reaction wood, knots, the slope of grain, fibre length, microfibril angle, extractives, durability, and permeability (for pressure treatment) [32,33].

Summing up, in the opinion of the authors, there is still not a standard method to assess loss of value due to aging in wood products, which is an important factor to know before investing in functionalization and stabilization techniques. The objective of this paper is to propose an objective loss of value measure so that the benefits of using higher quality wood materials can be compared with the, probable, higher cost. Instead of using a subjective definition of quality, this study proposes to use an objective monetary metric based on market price.

*Proposed Method*

It is argued in this paper that an objective measure of quality is the monetary value that customers are willing to pay for wood products. Even though market prices fluctuate by supply and demand, this fluctuation occurs within a range that reflects the perceived value of the products. Prices are not only a reflection of performance, but also customer satisfaction, brand recognition, promotion strategies and so on, thus market price, especially for commodities, provides a dynamic average of many factors affecting perceived value.

Since wood characteristics can vary substantially between and within trees and boards, wood products for building applications are typically delivered as components in large numbers with similar but not identical characteristics within defined ranges. These ranges are defined by producers, which use quality control parameters to sort the components into classes. Classes are usually called "grades" in industry jargon, which can be regarded as different commercial products. Grades can categorize components by quality, such as "premium", or intended application, such as "structural" in the case of timber.

In theory, it would be possible to re-grade wood products after use to estimate the retained value. If an old product falls into a lower grade than a new one, then the price difference can be interpreted as loss of monetary value due to aging. However, re-grading wood products after service is likely to be impractical. An obvious problem is that wood products are coated, screwed, nailed, sawed, glued and mechanically damaged by use through the years, and such defects do not exist in new products. A second problem is that a complete re-grading of wood products will provide the total loss of value caused by all defects simultaneously, while the purpose of this study is to assess the value retained by stabilization and functionalization techniques that may target one specific aging problem but not the rest.

For example, if a wood stabilization technique is specifically designed to delay the occurrence of checks, then the question is how much extra price customers would pay for not having checks in the product. In other words, the method proposed in this paper does not calculate the value difference between a new and old product, but the loss of value in a product after been downgraded for having a selected defect. The proposed method measures the extra price that costumers are willing pay for not having a selected defect in a wood product. If the selected defect develops later in service, then that extra price could be considered lost even if the rest of the product remains in good conditions.

Since commodity prices fluctuate by supply and demand, this study proposes to calculate the loss of value (LOV%) as percentage of the value that a product has without the defect. The proposed LOV% is calculated with Equation (1), based on the difference of

monetary value ($) between a wood product without a selected defect and the same wood product downgraded because of the selected defect:

$$\text{LOV\%} = 100 \ \frac{\$(\text{without defect}) - \$(\text{with defect})}{\$(\text{without defect})} \tag{1}$$

where:

LOV%: Loss of value for having a selected defect
$(without defect): Value of a wood product without a selected defect
$(with defect): Value of the same product downgraded due to the selected defect

## 2. Materials and Methods

In this article, LOV% is calculated for timber products produced according to the New Zealand grading rules (NZS 3631:1988) [26]. These rules define 26 different wood products listed in Table 1, with a range of dimensions covering 17 lengths, 8 widths, and 6 thicknesses.

**Table 1.** Timber classification in New Zealand's grading rules (NZS 3631:1988).

| | Appearance | Structural | Cuttings | Other |
|---|---|---|---|---|
| Native Softwood | (1) Clears (2) Premium (3) Dressing | (1) Building | (1) No. 1 Cuttings (2) No. 2 Cuttings | (1) Box |
| Hardwood | (1) Clears (2) Premium (3) Dressing | (1) Engineering (2) Building | (1) No. 1 Cuttings (2) No. 2 Cuttings | (1) Box |
| Exotic Softwood | (1) Clears (2) Select A (3) Select B (4) Dressing (5) Merchantable | (1) Engineering (2) No.1 Framing (3) No.2 Framing | (1) No. 1 Cuttings (2) No. 2 Cuttings | (1) Box |

Where: Exotic: are wood species that did not originate in New Zealand but were brought by settlers from abroad (such as radiata pine); Native: are wood species that originated in New Zealand before settlers arrived.

Stiffness and strength are the main properties of structural timber [34], but they are not supposed to change dramatically with time unless there is damage or decay. The example in this paper focuses on quality control parameters that are more likely to affect the visual appearance of wood, such as for wood in the external building envelope, thus structural grades are not included. The estimated market price was determined in consultation with a local sawmill that produces exotic softwood products. The prices were normalized by dividing the price of a specific product by the price of the most expensive appearance grade. Normalized products prices are summarized in Table 2, and the selected quality control parameters that are more likely to affect the visual appearance of wood are summarized in Table 3.

**Table 2.** Relative monetary values ($) of exotic softwood grades used to calculate LOV%.

| Appearance Grades | Relative $ |
|---|---|
| Clears | 1.00 |
| Select A | 0.67 |
| Select B | 0.60 |
| Dressing | 0.55 |
| Merchantable | 0.62 |
| No.1 cuttings | 0.67 |
| No.2 cuttings | 0.55 |
| Box | 0.39 |

The selected quality control parameters that may change or manifest due to aging are summarized in Table 3.

**Table 3.** Selected quality control parameters that may change or manifest during service.

| Defect | Principal Parameter | Secondary Parameter |
|---|---|---|
| Bow | Gap | |
| Crook | Gap | |
| Cup | Gap | |
| Twist | Gap | |
| Surface checks | Length | Width |
| Checks in knots | Width | |
| Collapse | Yes/No | |
| Shakes | Length | Slope |
| Splits | Length | |
| Stain | Yes/No | |
| Resin streaks | Length | Width |

Where: Gap: is the maximum distance between a curved wood surface placed on top of a straight reference table; Slope: is the angle with respect to the product centreline in the direction of the length.

Table 3 is a short version of the defects selected from the NZS 3631:1988 code. As defined in the code, a defect in Table 3 may refer to a group of defects. For example, shakes can be "closed", "through" and "non-through", and those three are considered distinct defects in the code. Furthermore, a defect mentioned Table 3 may refer to a single feature or a cluster of features. For example, surface checks may have a different impact based on the number or the cumulative of their lengths.

Table 3 also shows principal and secondary parameters, as most defects in the NZS 3631:1988 code are described by two parameters or less (2, 1 or 0 in some cases when the presence of a defect is enough to downgrade the product). For example, surface checks are described as having a length and a width. Surface checks also have depth, but it is not measured in the code. A surface check that penetrates deeply into the wood is called split and it is measured as a distinct defect. The only exception is Kaikaka, which requires measuring width, depth, and separation between affected areas. However, Kaikaka is a form of early decay that only affects one native New Zealand species called Totara.

Based on the considerations discussed above, a database was implemented in Excel to calculate LOV% in New Zealand timber. Units of measurement are not consistent throughout the New Zealand timber grading rules. In some cases, the defects are defined based on measurable parameters, such as the percentage of an area, and in other cases they are defined based on subjective thresholds. Therefore, the database uses mathematical relationships to transform the grading rules into comparable units of measurement.

Figure 1 shows the methodology used for transforming grading rules into comparable units of measurements. The database uses two parameters (Value 1 and Value 2) to describe both single defects and clusters of defects. For single defects the database uses a third parameter (Number) to indicate the maximum number of single defects that are permitted in the same piece of timber.

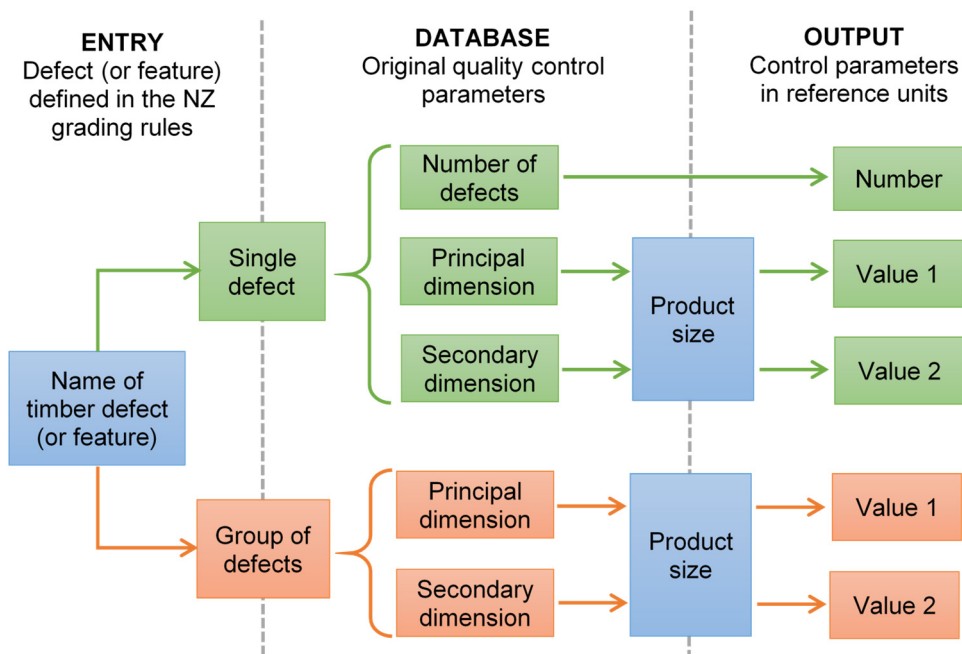

**Figure 1.** Methodology used to convert defects from the grading rules into comparable values.

The developed database produces an "Output" for each possible wood product and timber dimensions defined in the New Zealand grading rules. To calculate LOV%, the user enters a defect as Number, Value 1 and Value 2, and the database returns all possible quality sorts and timber dimensions that can accept such defect in the product.

## 3. Results and Discussion

For producing a numeric LOV% example, 5.1 m long by 150 mm wide by 50 mm thick exotic softwood timber was selected. Most of the timber defects listed in Table 3 are defined by threshold values ($\theta$). They produce one LOV% for defect sizes below $\theta$, and one LOV% for defect sizes above $\theta$. These results are summarized in Table 4, followed by the threshold values that apply to 5.1 m by 150 mm by 50 mm exotic softwood timber.

**Table 4.** LOV% for timber defects (X) characterized by a threshold value ($\theta$).

| Defect | Appearance Grades | |
|---|---|---|
| X | $0 < X < \theta$ | $X > \theta$ |
| Bow | 33% | 61% |
| Crook | 33% | 61% |
| Cup | 0 | 61% |
| Twist | 33% | 61% |
| Checks in knots | 38% | 38% |
| Shakes | 38% | 61% |
| Splits | 61% | 61% |
| Collapse | 33% | 38% |
| Stain | 33% | 61% |

Where: Bow $\theta$: 70.0 mm gap; Crook $\theta$: 25.0 mm gap; Cup $\theta$: 2.0 mm gap; Twist $\theta$: 12.5 mm gap; Checks in knots $\theta$: 2 mm wide; Shakes $\theta$: 1020 mm long and 3.8° angle; Splits $\theta$: any size; Collapse $\theta$: sufficient to affect the dressed dimensions; Stain $\theta$: sufficient to impair the natural finish.

The remaining defects that are not defined by thresholds are resin streaks and surface checks. These are defined by number, length, and width, thus there are many combinations of these dimensions that result in different LOV%. The analysis was simplified by comparing arbitrary lengths for selected widths and numbers.

LOV% due to resin streaks was calculated for width < 5 mm, 5 mm < width < 15 mm, 15 mm < width < 30 mm, and width > 30 mm. LOV% for surface checks was calculated for a single check with width < 0.5 mm, multiple checks with width < 0.5 mm, and multiple checks with width < 0.5 mm (where multiple means more than 3). The results are shown in Figures 2 and 3 respectively for resin streaks and surface checks.

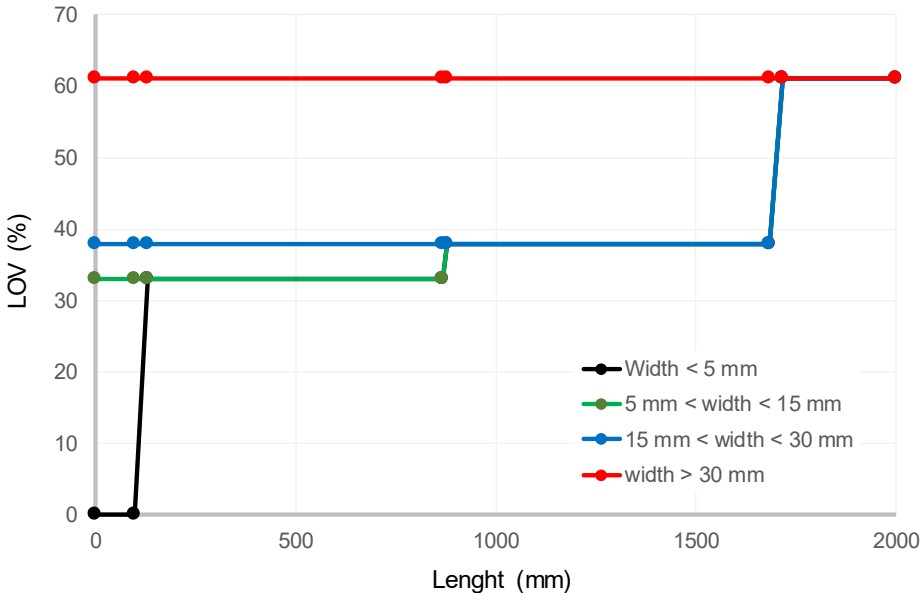

**Figure 2.** LOV% due to resin streaks in appearance grade timber.

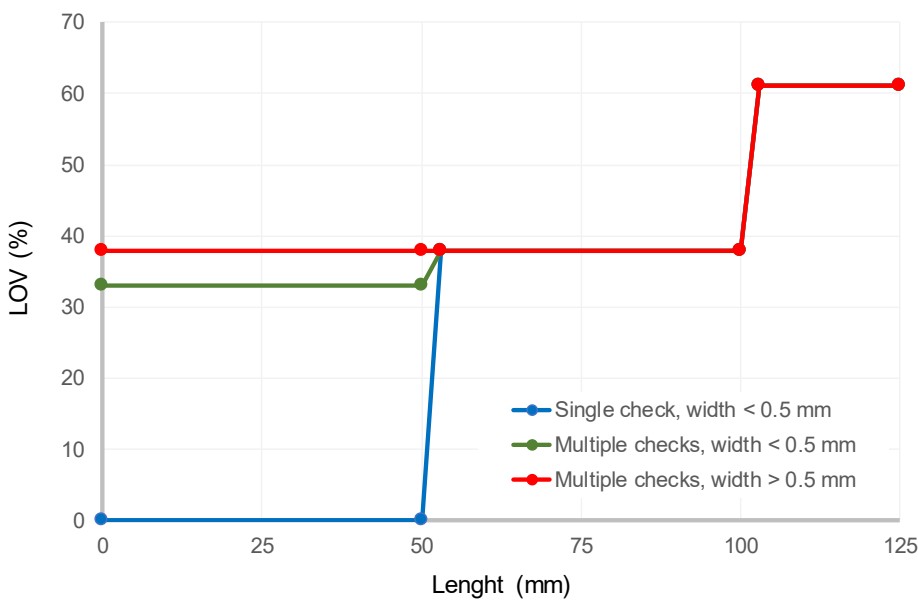

**Figure 3.** LOV% due to surface checks in appearance grade timber.

Calculated LOV% shows that exotic softwood timber in New Zealand can lose up to 61% of its value as appearance grade if a defect develops beyond the maximum allowed size. Furthermore, appearance grades can lose between 33% to 38% of their value by the mere presence of some defects regardless of the size. It must be said however, that the NZS 3631:1988 timber grading rules does not provide criteria for zero value timber. The lowest grade in the code is "Box", which still have commercial value. This limitation can be circumvented by looking at the applications. For example, if nobody uses Box

grade for external building façade, then Box grade will have inherently zero value for that application.

In addition, timber colour is not graded in the grading rules. The rules reject stain in timber when it is sufficient to impair the natural finish. But the natural finish of wood is expected to change in service when exposed to the external environment. Nevertheless, colour LOV% is difficult to quantify because most timber applications require protective coating, which typically mask the natural colour of timber. If other symptoms of aging are prevented by stabilization and functionalization techniques, then customers may find greying of wood acceptable, and even desirable if colour changes uniformly. A good example is wood shingles and shakes in houses, which are expected to become grey with time without detriment to appearance.

## 4. Conclusions

Wood quality is difficult to define because the features used to describe it depend on customer satisfaction, wood fibre characteristics, products, and applications. In addition, wood is not a product with specific purpose, but a material with many potential uses.

This discussion paper proposes an objective measure of loss of value due to wood defects that develop during aging. In short, the method consists in determining the maximum $ value that will be paid for a wood product that has a certain defect and compare with the maximum $ value that people will pay for a similar wood product that does not have such defect.

As an example, the paper uses the New Zealand timber grading rules to calculate loss of value in appearance softwood timber. It is argued that grading rules, both standards or stablished individually by producers, provide a practical metric for the balance between price and customer satisfaction in comparison with the alternatives.

The ultimate goal would be to generate databases containing information of quality control parameters for different timber products and applications around the world. Any person could then select a wood product and calculate LOV% due to aging defects, which will in turn determine the long-term economic benefit of applying stabilization and functionalization techniques.

**Author Contributions:** Conceptualization, D.E.; data curation, D.E.; formal analysis, D.E.; methodology, D.E.; project administration, D.G.; supervision, D.G.; writing—review and editing, D.E. and D.G. All authors have read and agreed to the published version of the manuscript.

**Funding:** This study was funded by New Zealand's Ministry of Business, Innovation, and Employment (MBIE), through The Strategic Science Investment Fund (SSIF).

**Institutional Review Board Statement:** Not applicable.

**Informed Consent Statement:** Not applicable.

**Data Availability Statement:** All data analysed in this study is publicly available from the New Zealand Timber Grading Rules NZS 3631:1988 [26].

**Conflicts of Interest:** The author of this study does not have any financial, commercial, legal, or professional relationship with other parties that could have influenced the research.

**Ethics Statements:** This material is the author's own original work, which has not been previously published or is currently being considered for publication elsewhere. The paper reflects the author's own analysis in a truthful and complete manner, in the context of prior and existing research, and having all sources properly disclosed.

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
