# Peer review of "A New Approach to Assess the Retained Value of Functionalized and Stabilized Wood Products through Aging"

_forests, doi:10.3390/f13050643_

Round 1
Reviewer 1 Report
With regard to the above mentioned manuscript, the topic is interesting and well-suited for the esteemed journal of Forests. The whole manuscript is clearly written and there is a logical flow of information throughout the text. However, there are some major concerns which I recommend being reconsidered by the authors as follows:
- Title: The Title clearly mentions “Stabilization and Functionalization Technique”. However, throughout the manuscript, there is no mention or comparison of different stabilization techniques carried out on wood specimens which are further compared with those wood specimens with no technique carried out on them. This has to be cleared and explained more in details to avoid any ambiguity. Further, some of the most important modification techniques are recommended to be briefly referenced in the Introduction section. In this regard, works by Prof. Holger Militz, Prof. Antonios Papadopoulos, and Prof. Mehdi Tajvidi will satisfy this area.
- Abstract: The last three lines starting with “Overall, the goal …” is better be repositioned somewhere in the 19th Line of Abstract before the sentence starting with “A numeric example …”. The normal position of “goals” in an Abstract is before declaring the Results. Moreover, the sentence in Line 20 starting with “The results showed, …” should be brought at the end of the Abstract.
- Abstract: All throughout the text, the idea of reduction of the quality of wood in service and aging is repeated. However, no mention of the time is made. I believe it should clearly be explained how the aging was measured, and the time that caused the defects in between. I believe the factor of time and duration is ignored.
- General: I believe the term “supply and demand” is more common internationally in comparison to “offer and demand”. So, I recommend modifying throughout the manuscript.
- General: While I approve the English writing, I might as well mention that a careful final editing should be done on the whole manuscript to correct some overlooked typing errors. Example: Line 216 “The results are shows respectively …”. (Shows should be corrected to shown.)
- General: Font size and format should be checked throughout the text. (Example: On the first line of Abstract, “Wood stabilization” is written with a larger size.)
- Introduction & References: While I quite approve all the references already cited in the manuscript, I would also emphasize that in the Introduction section, wood can be generally described and some general biological defects (even such as biological defects by wood deteriorating fungi, insects, and termites, and fire as well) should briefly be mentioned. Works by some eminent researchers (especially in forms of books and book chapters) are recommended to be added, like Prof. Mehdi Tajvidi, Prof. Ayoub Esmailpour, Mr. Jack Norton, Prof. Antonios Papadopoulos.
I believe that other parts are written and discussed very well.
Author Response
Hello,
Please find a file attached with the answers to the reviewers. There are three reviewers included, two from the peer-review process and one internal from Scion. Some questions were grouped in topics, so that questions are not answered two times. The colours used for the reviewers’ questions also match the colours used to highlight new text in the manuscript.
Scion’s reviewer also suggested to change the title, which has been change to the following in the new version of the manuscript: “A new approach to assess the retained value of functionalized and stabilized wood products through aging”
Best regards,
Diego Elustondo

Reviewer 2 Report
Please see the attached file.

Author Response

(The authors gave the same response as above.)
